# Establishment and Characterization of a Brca1^−/−^, p53^−/−^ Mouse Mammary Tumor Cell Line

**DOI:** 10.3390/ijms21041185

**Published:** 2020-02-11

**Authors:** Lilla Hámori, Gyöngyi Kudlik, Kornélia Szebényi, Nóra Kucsma, Bálint Szeder, Ádám Póti, Ferenc Uher, György Várady, Dávid Szüts, József Tóvári, András Füredi, Gergely Szakács

**Affiliations:** 1Institute of Enzymology, Research Centre for Natural Sciences, 1117 Budapest, Hungary; hamori.lilla@ttk.mta.hu (L.H.); kudlik.gyongyi@ttk.hu (G.K.); ks863@cam.ac.uk (K.S.); kucsma.nora@ttk.hu (N.K.); szeder.balint@ttk.hu (B.S.); poti.adam@ttk.hu (Á.P.); varady.gyorgy@ttk.hu (G.V.); szuts.david@ttk.hu (D.S.); 2Institute of Cancer Research, Medical University of Vienna, 1090 Vienna, Austria; 3Central Hospital of Southern Pest—National Institute of Hematology and Infectious Diseases, 1097 Budapest, Hungary; uher.ferenc@gmail.com; 4Department of Experimental Pharmacology, National Institute of Oncology, 1122, Budapest, Hungary; tozsi@oncol.hu

**Keywords:** breast cancer, BRCA1, cancer cell line, genetically engineered mouse model

## Abstract

Breast cancer is the most commonly occurring cancer in women and the second most common cancer overall. By the age of 80, the estimated risk for breast cancer for women with germline BRCA1 or BRCA2 mutations is around 80%. Genetically engineered BRCA1-deficient mouse models offer a unique opportunity to study the pathogenesis and therapy of triple negative breast cancer. Here we present a newly established Brca1^−/−^, p53^−/−^ mouse mammary tumor cell line, designated as CST. CST shows prominent features of BRCA1-mutated triple-negative breast cancers including increased motility, high proliferation rate, genome instability and sensitivity to platinum chemotherapy and PARP inhibitors (olaparib, veliparib, rucaparib and talazoparib). Genomic instability of CST cells was confirmed by whole genome sequencing, which also revealed the presence of COSMIC (Catalogue of Somatic Mutations in Cancer) mutation signatures 3 and 8 associated with homologous recombination (HR) deficiency. In vitro sensitivity of CST cells was tested against 11 chemotherapy agents. Tumors derived from orthotopically injected CST-mCherry cells in FVB-GFP mice showed sensitivity to cisplatin, providing a new model to study the cooperation of BRCA1-KO, mCherry-positive tumor cells and the GFP-expressing stromal compartment in therapy resistance and metastasis formation. In summary, we have established CST cells as a new model recapitulating major characteristics of BRCA1-negative breast cancers.

## 1. Introduction

Breast cancer is the most common cancer in women worldwide, affecting approximately one in eight women, and showing a heterogeneous population of tumors with different morphological and molecular features, prognoses and response to therapy [1,2]. In case of hormone- or growth factor-dependent breast cancers expressing ER (estrogen receptor), PR (progesterone receptor) and/or epidermal growth factor receptor 2 (HER2) targeted treatment options can significantly increase quality of life and survival. Conversely, triple negative breast cancer (TNBC), accounting for 15–20% of all breast cancer cases, carry a poor prognosis, as these tumors are insensitive to most available hormonal or targeted therapeutic agents [3,4,5].

Germline mutations in BRCA1 are responsible for a large proportion of inherited predispositions to breast and ovarian cancer. Women who carry mutations in BRCA1 or BRCA2 face 60% to 80% elevated lifetime risk to develop breast cancer by the age of 80 [6,7,8,9]. Germline mutations in BRCA1/2 occur in approximately 10% of TNBC patients [7], and mutations in further genes involved in the maintenance of genomic integrity and DNA repair, such as ATM (ataxia-telangiectasia), CHEK2 (Checkpoint kinase 2), and TP53 are also common [10,11]. BRCA-related mammary tumors are considered a distinct subtype due to their unique mutation profile (“BRCAness”) [12,13]. Breast cancers with BRCA1 or BRCA2 mutations are characterized by different gene expression patterns, highlighting the influence of heritable mutations on the phenotype and chemosensitivity of cancer [14,15].

In lack of targeted therapeutic options, chemotherapy remains the most effective treatment for TNBC. Neoadjuvant chemotherapy regimens containing taxanes and anthracyclines achieve complete response (CR) in 30% to 50% of patients with TNBC [16,17]. Unfortunately, TNBC patients with mutated BRCA1 are resistant to taxane therapy [18,19,20]. Conversely, drugs that induce DNA double-strand breaks and thereby increase genomic instability, represent a promising strategy for the treatment of TNBC with BRCA1/2 mutations [16,21,22]. Clinical studies have shown that platinum-based neoadjuvant chemotherapy is highly effective in TNBC patients with BRCA1 gene mutations [23,24]. Since 2018, patients relapsing after previous chemotherapy in the neoadjuvant, adjuvant, or metastatic setting are eligible to treatment with PARP inhibitors such as olaparib and talazoparib, inducing synthetic lethality in BRCA1 or BRCA2 cancers [21,25,26,27].

Preclinical cancer research has relied on animal models for the development of effective treatment for different cancer types. In particular, xenograft cancer models, where human cancer cells are transplanted in immunocompromised mice, offer an easy to use, inexpensive, and reproducible method. However, xenografts bear little resemblance with the molecular complexity and tumor heterogeneity of the original tumors; the vascularization of the heterotopic xenografts is often poor, and the stroma-tumor interactions in the immunocompromised hosts are very different from those influencing naturally occurring cancers in humans. For these reasons, while xenografts have been the most commonly used in vivo tumor models, their predictive value for therapeutic success has remained limited [28]. Genetically engineered mouse models of cancer (GEMMC) recapitulate molecular and histopathological features of the human disease and therefore provide a more sophisticated approach. GEMM of breast cancer captures at least some of the heterogeneity of human breast cancer [29]. Since the therapeutic efficacy of several anticancer agents relies on their capacity to influence the tumor-host interaction and anti-tumor immune responses [30], models with an intact immune system are important for the development of novel therapeutic strategies [29]. GEMMC have been successfully used for the preclinical evaluation of novel therapeutics, the study of the response of real tumors to therapy, and the identification of resistance mechanisms [31,32].

Over 20 distinct mutations, including null, hypomorphic, isoform, conditional, and point mutations, have been engineered in mice to study the relevance of Brca1 in mammary development and tumorigenesis [33]. Since germ-line deletion of Brca1 proved to be lethal during embryogenesis, and its heterozygous loss did not initiate tumorigenesis [34,35,36], development of genetically engineered mouse models of Brca1-related cancer required tissue-specific conditional knockout systems based on the cytokeratin 14 (CK14), β-lactoglobulin, MMTV-LTR or WAP driven expression of the Cre recombinase [37,38,39,40]. Loss of BRCA1 in epithelial tissues leads to mammary tumors, with long latency and low frequency [41,42]. Additional deletion of p53 significantly increases the incidence of breast cancer in these animals [43,44]. CK14-Cre driven somatic deletion of Brca1 and p53 resulted in solid carcinomas resembling high-grade IDC-NOS (invasive ductal carcinoma not otherwise specified) in humans. Significantly, these mammary tumors are highly proliferative, show ER-negativity and a high degree of genomic instability, similarly to human BRCA1-mutated hereditary breast cancers and sporadic basal-like breast cancers [37].

Over the years, the K14cre; Brca1^F/F^; p53^F/F^ mouse model of hereditary breast cancer has proved to be a useful tool to study tumor response and acquired therapy resistance of BRCA1-deficient breast cancers [45,46,47,48,49]. Brca1^−/−^, p53^−/−^ tumors show initial response to therapy, but eventually all tumors acquire resistance to doxorubicin, docetaxel [50] and olaparib [45] but not to cisplatin [50,51]. Preclinical studies in K14cre; Brca1^F/F^; p53^F/F^ mice revealed several mechanisms of resistance, such as elevated levels of drug efflux transporters and restoration of HR [45,46,52,53]. GEM models also possess limitations, as the synchronous overexpression or inactivation of potent oncogenes and tumor suppressor genes, respectively, often bypasses major bottlenecks to malignant transformation, resulting in reduced clonal heterogeneity [54,55]. Also, the generation of GEM models is expensive, and species differences must be carefully considered in experimental designs and interpretations.

Spontaneous tumors developing in K14cre; Brca1^F/F^; p53^F/F^ mice can be serially transplanted, offering a more convenient, cost effective and reproducible model for tumor intervention studies. Orthotopically transplanted tumor pieces give rise to tumors with the same basal-like phenotype, gene expression profile, initial sensitivity and acquired resistance to anticancer agents [37].

In vitro tumor models are widely used to study molecular mechanisms of tumor cell biology. Cell lines are important tools for cancer research and serve as low-cost screening platforms for drug development. Cells are easy to grow, and they provide an unlimited supply of material bypassing ethical concerns associated with the use of animal and human tissue [56].

In this study, we describe the phenotypic features and in vivo tumorigenicity of a new murine cell line derived from Brca1^−/−^; p53^−/−^ mammary tumors.

## 2. Results

### 2.1. Establishment of a Brca1^−/−^, p53^−/−^ Mouse Mammary Tumor Cell Line

Tumor pieces obtained from K14cre; Brca1^F/F^; p53^F/F^ mice [37] were orthotopically transplanted into the 4th mammary fat pad of female wild-type FVB mice [50]. When the tumors reached 200 mm^3^, the animals were sacrificed, and the tumors were removed. The tumor was cut into pieces, and following digestion with collagenase and dispase, the cells were seeded in primary culture medium as described in Materials and Methods. Initially, the tissue culture consisted of large fibroblast-like stromal cells and smaller, rapidly dividing cancer cells. To compare cancer cells to non-cancerous tissue derived from the same host, we isolated mesenchymal stem cells (MSC) from wild-type FVB mice (Figure 1A). As shown on Appendix A, the established cell line fulfills criteria commonly used for defining multipotent mouse mesenchymal stem/stromal cells, including adherence to plastic surface, specific cell-surface marker pattern and differentiation capability [57,58,59]. To favour cancer cells, horse serum, which is needed for the maintenance of primary fibroblasts, was removed. As a result, after approximately three passages (~3 weeks), fibroblasts disappeared from the flasks, and the primary cultures exhibited a uniform morphology corresponding to cancer cells (Figure 1B). During the following months, the initial morphology observed in the primary cell cultures gradually changed. The established cell line, designated as CST, consists of adherent cells exhibiting fillopodial and lamellipodial structures related to mesenchymal morphology (Figure 1C). Despite their mesenchymal phenotype, CST cells are of epithelial origin, as evidenced by the truncation of Brca1 gene that occurred in the mammary epithelium of K14cre; Brca1^F/F^; p53^F/F^ mice (Figure 1D).

### 2.2. Characterization of CST Cells

CST cells were kept in culture for up to 40 passages (~5 months) without any sign of senescence. Following expansion and cryopreservation at early passages, CST cells were further characterized in the context of epithelial and mesenchymal breast cancer cell lines (Figure 2). MCF7 cells (established from human Luminal A, ER+, PR+, HER2- breast cancer [60]) express high levels of the epithelial cell marker E-cadherin (also known as Cadherin-1). 4T1 cells, derived from a murine mammary carcinoma [61], exhibit epithelial morphology, E-cadherin expression, and also some degree of vimentin positivity. MDA-MB-231 is a human mesenchymal breast cancer cell line (triple-negative subtype: ER-, PR-, HER2-, claudin-low [60]), lacking E-cadherin expression and showing 100% vimentin positivity. As shown in Figure 2A, CST cells express vimentin, a prominent marker of the mesenchymal phenotype, whereas the expression of E-cadherin in CST cells is undetectable. Expression of cytokeratin 8 (CK8) and 14 (CK14) were also investigated (Appendix A). CST cells were positive for both, however, in contrast to the pattern revealed by immunohistochemical staining of tissue sections of K14cre; Brca1^F/F^; p53^F/F^ tumors [37], we observed nuclear localization of CK8 in CST cells. 

In vitro proliferation of CST cells was characterized by live-cell microscopy. Based on the assessment of relative confluency and cell numbers, the estimated doubling time of CST cells is 34 h, similarly to the well-characterized 4T1 cell line, whose doubling time is 22.9 h [62] (Figure 2B, Appendix A). Mesenchymal tumor cells are expected to exhibit increased motility and migratory capability. Cellular motility of the two mesenchymal cell lines were compared in a wound healing assay, with cell migration upon wounding assessed over a period of 24 h. As shown in Figure 2C, CST cells migrated within 24 h across the initial wound. By contrast, MDA-MB-231 cells exhibited lower levels of migration closing only 50% of the wound during the same time period (Appendix A).

Chromosomal integrity of the cell lines was characterized by quantifying γ-H2AX foci, which is an accepted measure of the number of DNA double-strand breaks in single cells. In line with the increased genomic instability of BRCA1-deleted cells [63], CST cells exhibited high numbers of spontaneous γ-H2AX foci. The high number of γ-H2AX foci in MDA-MB-231 is linked to the low constitutive expression of p21^WAF1^ (Figure 2D, Appendix A).

Genomic instability of CST cells was confirmed by whole genome sequencing, which revealed a very high number of single nucleotide variations (SNVs) compared to the FVB mouse genome, and frequent copy number changes indicating chromosomal instability. Genomic DNA was prepared from a single CST cell cloned population and sequenced on an Illumina HiSeq X Ten instrument with a mean coverage of 32. We identified 125,415 point mutations and 383,672 indels relative to the background mouse strain, of which 95156 and 268,669 were heterozygous and homozygous, respectively. A deconstruction of the SNV spectrum into COSMIC mutational signatures derived from cancer sequences detected the presence of mutation signatures 3 and 8 associated with HR deficiency [64], as well as signature 18, which was described in different cancer types including breast cancer, in association with oxidative damage [65] (Figure 3C). In addition to single base substitution signatures, we also predicted the copy number status in 16 kbp bins, which revealed significant copy number changes and genome-wide chromosomal instability (Figure 3B, Appendix A), corroborating our results of γ-H2AX quantification. The multiple copy number changes we detected on each chromosome indicated similar chromosomal instability as was observed in tumor samples from the same mouse model (Appendix A) [66].

### 2.3. In Vitro Chemosensitivity of CST Cells

In vitro sensitivity of CST cells was tested against 11 chemotherapy agents used in the treatment of breast cancers. Table 1 shows the IC_50_ values obtained with PrestoBlue^®^ assays, as well as the reference plasma concentrations of the chemotherapeutic agents (see also Appendix A). In general, CST cells show sensitivity to the tested drugs (IC_50_ values were between 0.02–1.2 µM). In line with the clinical efficiency of the tested drugs, CST cells are highly sensitive to SN38, which is the pharmacologically active metabolite of irinotecan; cisplatin, and the PARP inhibitors olaparib, veliparib, rucaparib and talazoparib.

### 2.4. Lentivirally Transduced CST Sublines Are Suitable to Study Tumor Formation, Anticancer Drug Response and Tumor-Stroma Interactions

Fluorescent protein expressing CST sublines were established by introducing either green fluorescent protein (GFP) or mCherry coding plasmids through lentiviral transduction (Appendix A). Stable expression of the fluorescent proteins did not change the phenotype of CST cells (Appendix A, Appendix A). To study the tumorigenic potential of the newly established CST lines, 1.5 × 10^6^ cells were orthotopically transplanted into the 4th fat pad of wild-type female FVB mice. CST cells proved to be tumorigenic in ~100% of the inoculated mice (*n* = 60). Mammary tumor formation was detected after 20 days (Figure 4A). Growth kinetics of the CST derived tumors were similar to the rates observed with the serial orthotopic transplantation of tumor pieces [45]. Tumor formation potential of the CST lines expressing GFP or mCherry was also evaluated. 25 days after inoculation, CST-mCherry tumors became apparent and continued to grow until the experimental endpoint (Figure 4B).

Tumors derived from orthotopically transplanted tumor pieces show sensitivity to cisplatin [50]. To test the in vivo drug response of CST cells, 1.5 × 10^6^ CST-mCherry cells were orthotopically injected into FVB-GFP mice (FVB.Cg-Tg(CAG-EGFP) B5Nagy/J). When the tumors reached 200 mm^3^, mice were treated with the maximum tolerable dose (6mg/kg) of cisplatin with 2-week intervals. Similarly to results obtained with orthotopically transplanted tumor pieces, CST-derived tumors responded well to cisplatin, relapsing tumors remained sensitive to cisplatin, but the tumors were not eradicated (Figure 4C).

The fluorescence of CST cells offers a tool to investigate tumor-stroma interactions. To allow efficient separation of tumor and stroma cells, 1.5 × 10^6^ CST-mCherry cells were orthotopically injected into GFP-positive FVB mice. When the tumors reached 200 mm^3^, the animals were sacrificed, and the tumors were removed. Following digestion with collagenase and dispase, the cells were seeded in primary culture medium as described in Materials and Methods. In these primary cultures, GFP-positive host fibroblast cells form nests in the midst of cancer cells expressing mCherry (Figure 4D). Next, the cells were sorted based on mCherry/GFP expression, and sorted cells were cultured separately. As shown in Figure 4E, mCherry-positive CST cells preserved the characteristic mesenchymal morphology, while GFP-positive fibroblasts are larger, and exhibit a flat, polygonal, stellate-like morphology with formed lamellipodia.

## 3. Discussion

Whereas tumors grow vigorously in vivo, continuously bypassing cellular obstacles such as cell cycle regulation or apoptosis, the establishment of cancer cell lines is not a straightforward process. In vitro, cells have to adapt to the lack of the original microenvironment consisting of stromal and immune cells, the different oxygen levels, the diverse composition of growth factors, and they have to adhere to the plastic surface of the tissue flask. Due to changes occurring during the in vitro adaptation process, stabilization and characterization of a new cell line should follow published guidelines, demonstrating the immortality, neoplasticity, origin, contamination-free background and scientific significance of the newly established cell line [70,71].

Here we present a new murine cancer cell line, designated as CST, which was isolated from a genetically engineered Brca1^−/−^; p53^−/−^ mouse model of hereditary breast cancer. The role of BRCA1 in familial breast and ovarian cancer is well documented [72,73], but in recent years its importance in several other malignancies has been also discovered. Mutations in the Brca1 gene were found in urothelial tumors [74], pancreatic cancer [75], and prostate malignancies [76].

CST cells show mesenchymal morphology and express mesenchymal markers; and the rapid growth rate and increased motility also indicate an aggressive phenotype, a characteristic of the original mouse tumor as well, from which CST cells were derived [37]. CST cells share several features with BRCA1-mutant human breast cancer cell lines (HCC1937, MDA-MB-436, SUM1315MO2, and SUM149PT [77,78,79]. For example, the BRCA1-mutation was shown to deregulate the expression of E-cadherin [78], keratin 8 and vimentin [80], and reduced BRCA1 function often results in increased proliferation, cellular migration and invasion [81]. These features correspond to human BRCA1-deficient basal-like breast cancers, which are characterized by high proliferation index, triple-negative subtype, high degree of genomic instability and expression of basal epithelial markers [37]. Patients with basal-like tumors have poor outcome, short survival and relatively high mortality rate [82].

The extensive DNA damage observed in untreated CST cells (Figure 2D) is in line with the defective homologous recombination due to loss of BRCA1 [83]. This is also supported by whole genome sequencing: both the presence of COSMIC signatures 3 and 8 and the elevated levels of copy number changes are characteristic of reduced homologous recombination in cells lacking BRCA1 (Figure 3). As a result, CST cells show high sensitivity to DNA damaging compounds (Table 1). In particular, topoisomerase inhibitors (doxorubicin, etoposide, SN38 and epirubicin) strongly inhibit the growth and viability of CST cells, with IC_50_ values around 100 nM, corresponding to the in vitro, in vivo and clinical hypersensitivity of BRCA1-deficient tumors to anthracyclines [49,84,85,86]. In contrast, cells with mutated BRCA1 usually show increased tolerance to microtubule targeting drugs, which is explained by the role of BRCA1 in mitotic-spindle assembly and the JNK pathway linking damage of the microtubule network to apoptosis [87,88,89]. CST cells are about 10-fold resistant to paclitaxel compared to most BRCA1-proficient breast cancer cell lines [90,91]. On the other hand, impaired homologous recombination due to dysfunctional BRCA1 creates a targetable dependency on Poly (ADP-ribose) polymerase (PARP) activity and sensitizes cells to platinum drugs. As expected, CST cells are sensitive to cisplatin, a first-line platinum chemotherapy drug in BRCA1-mutated breast tumors, and to a panel of PARP inhibitors (olaparib, veliparib, rucaparib and talazoparib) designed to exploit the synthetic lethality of BRCA1-linked cancers.

While the cellular consequences of BRCA1-mutations are well characterized, the interaction between cancer cells lacking BRCA1 and their microenvironment is less studied. To establish in vitro and in vivo models for the study of the cooperation between BRCA1-KO tumor cells and the stromal compartment, we transduced CST cells with lentiviral plasmids encoding GFP (CST-GFP) or mCherry (CST-mCh). Since stable expression of the fluorescent proteins did not change the phenotype of CST cells (Appendix A), in vivo tumor formation and chemosensitivity was evaluated in GFP-positive FVB mice orthotopically engrafted with CST-mCh cells. This experimental model gave rise to mammary tumors containing mCh-positive cancer cells and GFP-positive stromal cells (Figure 4). Primary cultures established from these tumors revealed GFP-expressing nests of fibroblasts surrounding mCherry-positive cancer cells. Following sorting by FACS, mCherry expressing CST cells could be cultured separately from GFP-positive stromal cells showing the usual characteristics of fibroblast/mesenchymal stem cells. Dual coloring of cancerous and healthy cells will allow further experiments addressing the complexity of the tumor microenvironment [92]. Using a similar approach, Kidd et al. [92] showed that while fibroblast/mesenchymal stem cells originate from the bone marrow, vascular and fibrovascular support is derived from the local fat tissue. The widely used xenograft and allograft models of cancer rarely go into remission after treatment, usually only changes in the initial growth rates are observed. In contrast, the dually-labeled tumors derived from CST cells and host stroma show complete response to cisplatin treatment, a first-line therapy of BRCA1-mutated breast cancers. (Figure 4). Future studies will address the role of the stromal compartment in drug response, therapy resistance and relapse.

In summary, we have established CST cells as a new in vitro model recapitulating major characteristics of BRCA1-negative breast cancers. Thus, the CST cell line could become an important tool for the research of BRCA1-mutated cancers.

## 4. Materials and Methods

### 4.1. Isolation of Tumor Cells, Maintenance of Cell Lines

Tumor pieces obtained from K14cre; Brca1^F/F^; p53^F/F^ mice (a kind gift from Sven Rottenberg, University of Bern) were orthotopically transplanted into the 4th mammary fat pad of female wild-type FVB mice [50]. The original tumor model was established in FVB mice [93], therefore, all in vivo experiments were performed with FVB mice. When the tumors reached 200 mm^3^, the animals were sacrificed, and the tumors were removed. The tumor tissue was cut into pieces, washed with 1 X PBS and was digested with collagenase (200 U/mL) and dispase (0.6 U/mL) in DMEM for 30 min at 37 °C in a 50 mL Falcon tube while vortexed for 1 min every 10 min. The suspension was filtered through a 40 μm cell strainer to remove remaining extracellular matrix and the collected cell suspension was centrifuged. Cells were resuspended using primary culture medium (DMEM/F12  +  10% FBS +  10 % horse serum + 1% penicillin–streptomycin) and transferred to a culture flask overnight in a humidified incubator at 37 °C with 5% CO_2_. Mesenchymal stem cells were isolated from the bone marrow of FVB mice. Isolation and characterization of these cells were carried out as described earlier [57,94].

4T1 (a kind gift from Ágnes Csiszár from the Institute of Cancer Research, Medical University of Vienna) and CST cell lines were cultured in Dulbecco’s Modified Eagle Medium Nutrient Mixture F-12 (Thermo Fisher Scientific, Massachusetts, USA) supplemented with 10% FBS and 50 units/mL penicillin and streptomycin (Life Technologies). MDA-MB-231, MCF7 were obtained from the National Cancer Institute’s Developmental Therapeutics Program (National Institutes of Health) and were cultured in Roswell Park Memorial Institute (RPMI) 1640 Medium supplemented with 10% FBS, 5 mM glutamine, and 50 units/mL penicillin and streptomycin (Thermo Fisher Scientific, Massachusetts, USA). All cell lines were cultivated at 37 °C with 5% CO_2_.

### 4.2. Genotyping Brca1-Deleted Allele by PCR

Genotyping of Brca1^-^ allele was performed as described by Liu et al. [33]. Detection of Brca1-deleted allele with primers P1 and P4 yielded a 594-bp fragment. Wild-type Brca1 was detected by the product of P1-P2 primers (390 bp). The primer sequences were as follows: P1, 5′-TAT CAC CAC TGA ATC TCT ACC G -3′; P2, 5′-GAC CTC AAA CTC TGA GAT CCA C -3′; and P4, 5′-TCC ATA GCA TCT CCT TCT AAA C -3′. For all PCR reactions, thermocycling conditions consisted of 30 cycles of 30 sec at 94 °C, 30 sec at 58 °C, and 50 sec at 72 °C. Reactions contained 100 ng of template DNA, 0.4 uM primers, 100 mM dNTPs, 2.5 units of TaqDNA polymerase, 2.5 mM MgCl_2_, and 10 x PCR buffer in a 20-mL volume. After the electrophoresis of PCR products on 1% agarose gel, the gel was stained with ethidium bromide.

### 4.3. Live-Cell Microscopy, Growth and Motility Assay

Doubling time (T_d_) was obtained using the following formula: T_d_ = ln2/K, where K is the constant rate calculated from N_t_ = N_o_ × e^K^ × ^t^ where N_t_ is the cell number at time t; N_o_ is the cell number at the initial time point. The cell number was determined using automated cell counter every day for 4 days. To evaluate growth rates, cells were plated at a density of 2 × 10^4^ cells per well in 24-well culture plates. Images were taken every 12 h for 3 days. For the motility assay, cells were seeded in a 6-well plate at 10^6^ cells/well density. The wound was created by scratching a confluent monolayer with a 200 µL pipette tip, and wound healing was followed for 48 h by taking images every 6 h. All images were taken using the JuLI Stage Real-Time Cell History Recorder (NanoEnTek, Seoul, Korea) with a bright channel, 4x/0.16 U Plan S-Apo objective. Wound closure was quantified using the ImageJ plugin MRI Wound Healing tool (Volker Baecker, Montpellier RIO Imaging, Montpellier, France).

### 4.4. Drugs

Cytotoxic drugs were purchased directly from the manufacturers. The compounds used in the CST cytotoxicity assays were purchased from Accord Healthcare (cisplatin, epirubicin), Selleckchem (olaparib), Sigma-Aldrich (paclitaxel, SN-38, doxorubicin), TEVA (etoposide), Tocris (veliparib), Merk (gemcitabine) and MedChemExpress (talazoparib, rucaparib).

### 4.5. In Vitro Cytotoxicity Assay

Viability was assessed using the PrestoBlue^®^ assay (Life Technologies), according to the manufacturer’s instructions. Briefly, cells were plated in 96-well plates and treated in the given concentration range with the indicated compounds for 120 h. Viability of the cells was measured spectrophotometrically using an EnSpire microplate reader (Perkin Elmer). Data were normalized to untreated cells; curves were fitted by the GraphPad Prism 5 software (GraphPad, San Diego, California, USA) using the sigmoidal dose–response model. Curve fit statistics were used to determine mean concentrations of the selected drugs required to inhibit cell proliferation by 50% (IC_50_).

### 4.6. Lentiviral Transfection

2nd generation lentiviral vectors and packaging plasmids (pMD2.G and psPAX2) were obtained from Addgene. CST cells were transduced with GFP (pRRL-EF1-eGFP-WPRE) or mCherry (pRRL-EF1-mCherry-WPRE) expressing lentiviral supernatants prepared as follows: lentiviral particles were produced in HEK293T cells transfected by the calcium phosphate co-precipitation method [95]. On the starting day, 5 × 10^6^ HEK293T cells were plated in a 10 cm^2^ Petri dish. Calcium phosphate transfection was performed the following day. 24 h later, the medium was replaced with fresh medium, supplemented with KnockOut serum replacement (Gibco). The supernatant, containing the lentiviral particles, was harvested 48 h after transfection, filtered through a 0.45 μm syringe filter, and then stored at −80 °C until further use. The multiplicity of Infection was determined by flow cytometry. Transduction of target CST cells was carried out on 6 well plates. After the transduction, cell lines were tested and sorted by flow cytometry.

### 4.7. Immunocytochemistry and γ-H2AX Foci Number Analysis

Cells were seeded into 8-well μ-Slides (Ibidi) at a density of 80,000 cells/well. After overnight incubation, cells were washed with pre-warmed PBS, fixed with 4% formaldehyde solution for 15 min at room temperature, washed, and then blocked with complete blocking solution (0.5% BSA, 0.1% TritonX-100, 5% goat serum in sterile PBS) for one hour at room temperature. Next, samples were incubated overnight at 4 °C with the relevant primary antibodies (anti-E-Cadherin antibody (ab11512-Abcam), anti-Vimentin Antibody (V9) (sc-6260-SantaCruz), anti-gamma H2A.X (phospho S139) antibody (ab11174-Abcam) anti-cytokeratin 8 (ab53280-Abcam) and anti-cytokeratin 14 (ab7800-Abcam). After incubation, the cells were washed with PBS, and the secondary antibodies (Alexa Flour 488, Alexa Fluor 546, Alexa Fluor 555) were added in complete blocking solution, followed by incubation for two hours at room temperature. Nuclei were labeled with DAPI. Imaging was carried out using a ZEISS LSM-710 system (Carl Zeiss microscopy Gmbh, Jena, Germany) with a 40×/1.4 Plan-Apochromat oil immersion objective. Images were processed with ZEN (Carl Zeiss microscopy Gmbh, Jena, Germany). γ-H2AX foci were counted with FindFoci, an automated ImageJ plugin [96].

### 4.8. Animal Experiments

All animal protocols were approved by the Hungarian Animal Health and Animal Welfare Directorate according to the EU’s most recent directives. All surgical procedures were performed according to the Committee on the Care and Use of Laboratory Animals of the Council on Animal Care at the Institute of Enzymology, RCNS in Budapest, Hungary (001/2574–6/2015). Tumor cells (CST or CST-mCherry cells) (1.5 × 10^6^/animal) were injected into the mammary fat pad of 6–8 weeks old female FVB or GFP expressing FVB mice. The tumor size was monitored at least 3 times per week by caliper measurements after the tumors became palpable. Tumor volume was calculated using the V = length × (width2/2) formula. When the tumors volume reached 200 mm^3^, mammary tumors were either removed (control group) or treated with maximum tolerable dose of cisplatin (MTD, 6 mg/kg iv. respectively). Treatments using the MTD were repeated every 14 days. Animals were sacrificed when the tumor volume reached~2000 mm^3^.

### 4.9. Cell Sorting

Tumors were removed from the untreated control group. Removed tumors pieces were digested and filtered as described above. Sorting was performed using a FACSAria III cell sorter (BD Biosciences, San Jose, California, US) equipped with four lasers. EGFP fluorescence was measured using blue laser excitation (488 nm) and 530/30 nm emission (FITC-A channel); mCherry fluorescence was measured using yellow-green laser excitation (561 nm) and 610/20 nm emission (PE-mCherry channel). Flow cytometry data was collected and analysis was performed using FACSDiva 8.02 software.

### 4.10. Whole Genome Sequencing and Genomic Data Analysis

Whole genome sequencing of a single cell CST clone was done at Novogene, Beijing, China. The sequencing reads were quality controlled by FastQC [97], and filtered for bad quality and Illumina adapter containing sequences by Trimmomatic [98]. Sequence alignment was conducted using BWA MEM [99] against the GRCm38.p6 reference genome. Mutations were called by the GATK pipeline [100] using HaplotypeCaller with the mouse dbSNP142 for positive training. Single nucleotide variants were filtered for a minimum coverage of 10 and categorized according to COSMIC classification scheme. Heterozygous and homozygous spectra were separately refitted to the 30 COSMIC v2 mutational signatures using the deconstructSigs R package [101] with default settings. Large scale copy number levels in 16kb bins were estimated by goleft indexcov [102] and the ploidy was predicted by piecewise constant fitting using the rpart R package [103]. Copy number profiles of K14cre; Brca1^F/F^; p53^F/F^ mouse tumors [66] were downloaded from GEO accession GSE122076; ploidy levels were predicted by piecewise constant fitting of aCGH values, and positive and negative ploidy levels were averaged for each probe separately.

Whole genome sequence data obtained in this project is available from the European Nucleotide Archive under study accession number PRJEB36418.

## Figures and Tables

**Figure 1 ijms-21-01185-f001:**
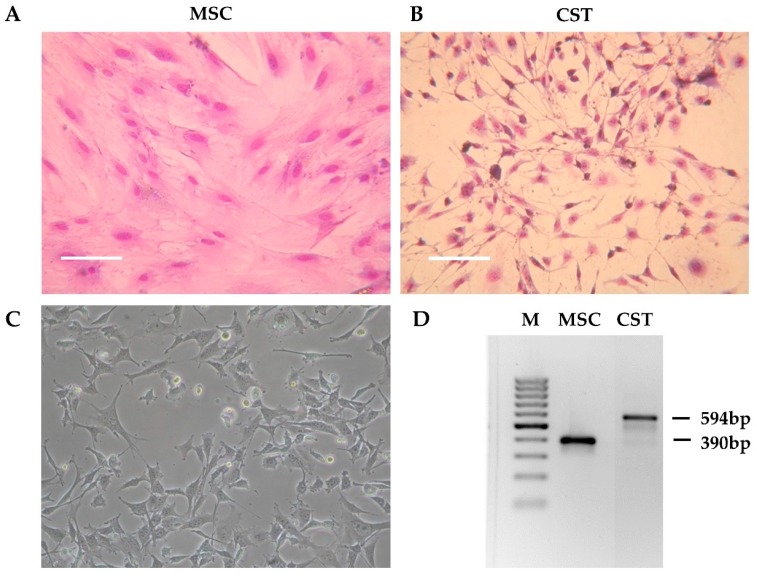
Preliminary characteristics of the CST cell line. (**A**) Morphology of mesenchymal stem cells (MSC) derived from the bone marrow of wild-type FVB mice. (**B**) Primary Brca1^−/−^, p53^−/−^ tumor cells after 3 weeks in culture (CST) (Giemsa staining). Digital photos were taken by a Nikon Coolpix 4500 digital camera (Nikon GmbH, Düsseldorf, Germany) connected to an Olympus CK2 inverted microscope (Olympus, Tokio, Japan) with 10x objective. (**C**) Established CST cell line (representative image). Microscopy pictures were acquired using 10x/0.25 Plan-Fluor objective with an Eclipse TS100 Inverted Microscope (Nikon, Japan). (**D**) PCR-based genotyping of mesenchymal stem cells (MSC) and CST cells (M—marker). The Brca1-deleted allele yields a 594-bp fragment, whereas wild-type Brca1 is detected by a shorter product (390 bp).

**Figure 2 ijms-21-01185-f002:**
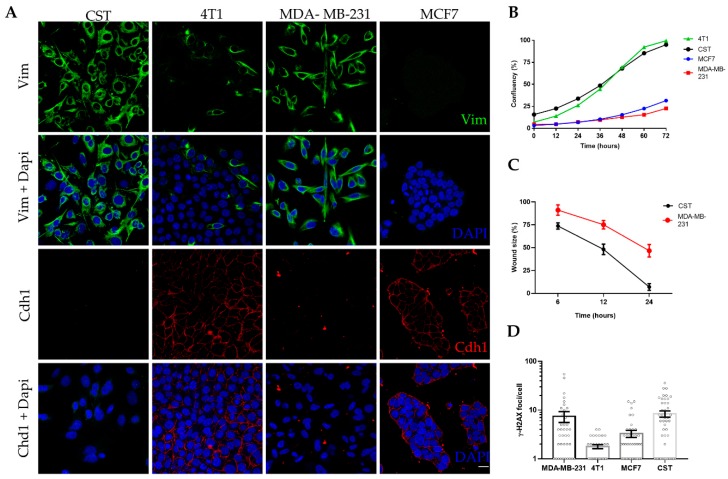
Characteristics of the CST cell line. (**A**) Immunofluorescent detection of vimentin (Vim) (green) and Cadherin-1 (Cdh1) (red); nuclei were stained with Dapi (blue), microscopy pictures were acquired using ZEISS LSM-710 system (Carl Zeiss microscopy Gmbh, Jena, Germany) with 40x/1.4 Plan-Apochromat oil immersion objective (images were processed with ZEN (Carl Zeiss microscopy Gmbh, Jena, Germany)). Scale bar = 20 µm. (**B**) Cell growth curves of four breast cancer cell lines. Confluency (%) was determined by live cell microscopy, with a JuLI Stage Real-Time Cell History Recorder (NanoEnTek) in bright field. (**C**) Wound healing assay performed with CST and MDA-MB-231 cells. Relative wound size (compared to initial size at the start of the experiment) is shown. Data points represent mean ± SD of 3 experiments. (**D**) Mean number of γ-H2AX foci per cell. Data represent mean ± SEM of γ-H2AX loci per cell nucleus; foci were counted with FindFoci, an automated ImageJ plugin, based on confocal images shown in Appendix A.

**Figure 3 ijms-21-01185-f003:**
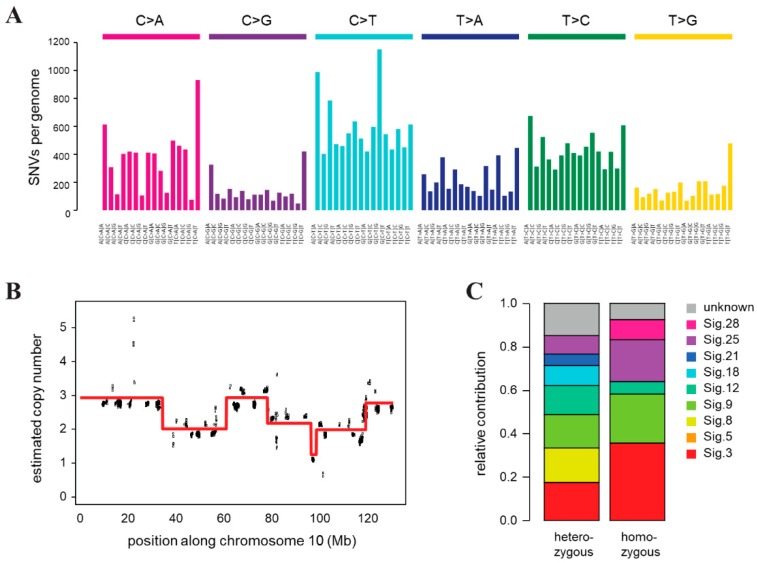
Genomic features of the CST cell line. (**A**) Triplet base substitution spectra of heterozygous mutations in CST. Base changes are summed according to the preceding and following positions, as shown below the panel. All positions marked as SNPs in the FVB mouse genome in dbSNP are filtered out. (**B**) An example of the copy number variations and ongoing chromosomal instability in the CST genome (chromosome 10 only). Estimated coverages in 16 kb windows (black) and the associated ploidy level predictions (red) are shown. (**C**) Decomposition of the spectrum of heterozygous and homozygous SNVs into COSMIC mutational signatures (sig.).

**Figure 4 ijms-21-01185-f004:**
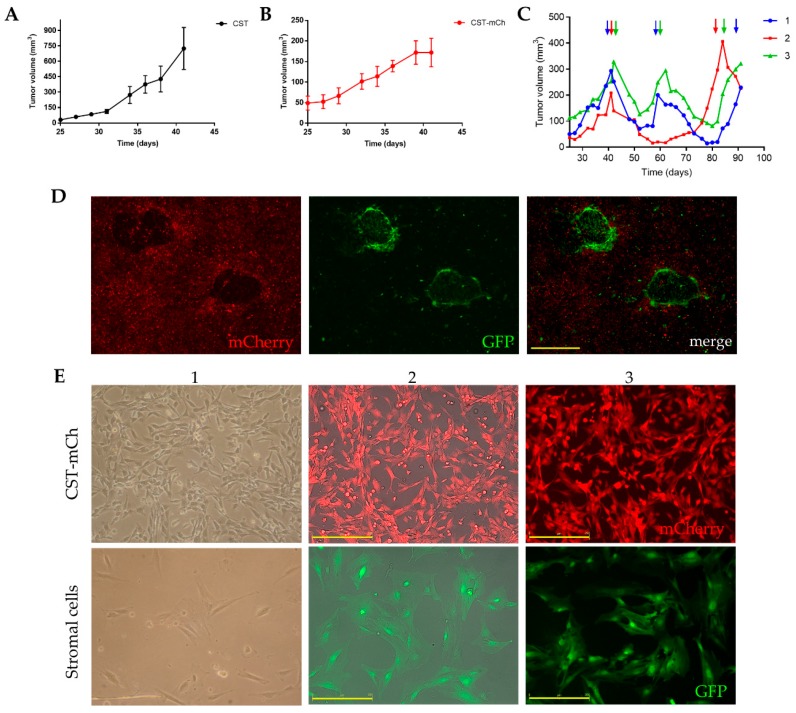
Lentivirally transduced fluorescent CST sublines offer a model system to study tumor formation, anticancer drug response and tumor-stroma interactions. (**A**) Growth kinetics of tumors derived from CST cells (1.5 × 10^6^ cells/mouse). Data represent mean tumor volumes ± SEM (*n* = 5). (**B**) Growth kinetics of tumors derived from mCherry expressing CST cells (1.5 × 10^6^ cells/mouse). Data represent mean tumor volumes ± SEM (*n* = 5). (**C**) Cisplatin treatment of orthotopically injected CST-mCherry tumor cells into GFP-expressing FVB mice. When the tumors reached 200 mm^3^, cisplatin was administered at the maximum tolerable dose (6mg/kg) as indicated by the arrows. (**D**) Primary culture of CST-mCherry derived tumor cells containing GFP-positive host cells. Scale bar = 500 µm. (**E**) Cultures of sorted mCherry-positive CST cells and GFP-positive stromal cells. 1—light microscopy 2—JuLi Stage bright field, RFP merge, 3-JuLi Stage RFP. Scale bar = 250 µm. Microscopy pictures were either acquired using JuLi™ Stage (NanoEnTek Inc., Korea) with 4x/0.16 U Plan S-Apo objective (Figure 4D), 10x/0.3 U Plan FLN objective (E2, E3) or using Nikon Eclipse TS100 Inverted Microscope (Nikon, Japan) with 10x/0.25 Plan-Fluor objective (E1).

**Table 1 ijms-21-01185-t001:** IC_50_ values and maximum plasma concentrations (C_max_) of the tested chemotherapeutic agents [67,68,69].

Chemotherapeutic Agents	IC_50_ (µM)	C_max_ (µM)
Doxorubicin	0.2	6.73 [67]
Cisplatin	1.1	14.40 [67]
Epirubicin	0.1	16.60 [67]
Paclitaxel	1.2	4.24 [67]
Etoposide	0.2	33.40 [67]
SN38	0.01	0.14 [67]
Gemcitabine	0.1	89.30 [67]
Olaparib	0.5	13.10 [67]
Veliparib	4.4	7.04 [68]
Rucaparib	0.2	6.00 [67]
Talazoparib	0.007	0.036 [69]

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
