# Peer review of "Establishment and Characterization of a Brca1−/−, p53−/− Mouse Mammary Tumor Cell Line"

_ijms, 2020, doi:10.3390/ijms21041185_

Round 1

Reviewer 1 Report

The authors isolated and analysed a cell line from a transplanted K14 Brca1;p53 mammary tumour. Additional analyses might add value to the cell model described here.

In addition to the analysis of E-caderin and Vimentin expression, analysis of keratin expression (pan-keratin and/or specific keratin types) might reveal a closer similarity of the CST cell line to human BRCA1-mutated tumours. Comparison of CST cells to human BRCA1-deficient breast cancer cells (i.e. HCC 1937 or others) would be more appropriate. Hollern D. P. et al. (Breast Cancer Research and Treatment, 2019) recently reported the genomic profile of a K14 Brca1;p53 mouse tumour and compared it to that of human basal breast cancers. Could the authors highlight any similarities between those sequencing data and the ones obtained from the analysis of CST cells ? 

Minor comments:

line 20: 'women' instead of 'woman' line 50: Did the authors mean 'CHK2' instead of 'CHEK2'? line 87: IDC-NST instead of IDC-NOS line 301: please, specify which type of collegians has been used.

Reviewer 2 Report

Overall, the work seems valuable and the new cell line could be useful for the studies on BRCA1-mutated breast tumors. In particular, syngeneic mice model constructed using these cells can be used e.g. for the studies on immunotherapy.

However, there is lack of information through the whole manuscript whether CST cell line is immortal and how many passages were done. 

Please provide the information whether CST cells are tumorigenic  in 100% of inoculated mice.

The results are poorly described in some sections, especially descriptions of figures need to be improved. Titles of the figures are inadequate in some cases. Figures are designed chaotically.

Quality of some images is poor.

Specific comments:

Lane 28: COSMIC abbreviation should be explain

It is worth to mention in the Introduction that BRCA1 mutation-related breast cancers show distinct gene expression profiles as determined by DNA microarray analyses, eg.

https://www.ncbi.nlm.nih.gov/pubmed/11207349

https://www.ncbi.nlm.nih.gov/pubmed/21196292

https://www.ncbi.nlm.nih.gov/pubmed/23704984

Lane 130: Mesenchymal stem cells used in this figure are not described anywhere (in Methods, in the figure description, etc.) What is the origin of the cells? How did you check that cells presented on Fig. 1A image (1) are (mesenchymal) stem cells?

Figure 1: the title is not adequate. It could be rather: “Preliminary characteristic of CST cell line”.

There is also no information in description of figure about the microscope and magnification used.

Description of images in part A (1, 2) is misleading: it looks like lanes in part C numbered 1, 2 and 3 contained material from cell cultures 1 and 2.

Images in part A should be better described by appropriate phrases (eg. CST cells instead of “2”)

Part B) lacks any description,

Part C) lanes could be described as: “M” (Marker), mesenchymal cells, CST cells. There is no explanation for the empty lines?

Line 149: should be “two mesenchymal cell lines” instead of “two mesenchymal cells”

Figure 2 is chaotic. It could be better organized and better described. The title is inadequate. It could be: Characteristics of CST cell line (cont.)

Descriptions of columns of images in part A) should be improved, for example, from first column on top: Vimentin (mesenchymal), Vim + Dapi (merge), Cdh1 (epithelial), Cdh1 + Dapi (merge)

In part D) – please re-scale y-axis (enlarge it in the range between 0-20) to improve legibility of the graphs

Figure 3 part A) Results of whole genome sequencing should be more accurately described (please explain that each mutation class has 16 categories and how they are identified), describe y axis, in C) describe better (add information that there are COSMIC signatures shown. Lane 182 - Change in the figure footer: Y= relative… for: y axis represents…

Lane 193-194 Description of Table 1 should be above the table.

207-212 lines: There is discrepancy between the information given here and that what is visible on the graph on Figure 4C, in case of tumor no 3 it seems that after second treatment there is lack of cisplatin response. Tumor no 2 seems to be almost eradicated after first treatment. Only tumor no 1 behavior corresponds to the description given in the text.

Figure 4E – please ensure good quality images for this part of the Figure. In addition, a scale bar is poorly visible.

Figure 4 is chaotically designed. Microscopic images should be larger

There is no reference for Fig. S5 in main text.

Lane 289 – in this place Authors write about complete response of the tumors to cisplatin treatment what is in contrast to the statement in lane 207-212 and what is seen on the Figure 4.

Lane 296: please give an information on what genetic background are these mice constructed

Chapter 4.1 – please give an information about mesenchymal stem cells used in Fig. 1

Lane 307 -  Agnes Csiszar FROM THE Institute

Round 2

Reviewer 1 Report

Please, briefly describe the results of the staining for CK14 and CK8 in CST cells in the main text. Also, clearly state that CK8 expression does not appear to mirror the pattern observed in K14 Brca1 p53 tumours (described in Liu X., et al, 2007)

Reviewer 2 Report

Big thanks to Authors for their effort to adhere to the Reviewers comments. In my opinion the manuscript has been significantly improved.

I still have two minor notes.

There is an ambiguity between main text (lanes 127-140) and Figure 1.

For greater clarity I propose the change in description of images in Figure 1:

Please split  present part A into two parts (A and B), and rename present B to C, present part C to D Descriptions of the images could be as follows:

A Bone marrow MSC

B CST after 3 weeks culture

C Established CST cell line

D leave present description without changes

I propose following references within the text pertaining to Figure 1:

Line 128 – remove reference to Fig. 1A

Line 128-130: To compare cancer cells to non-cancerous tissue derived from the same host, we isolated mesenchymal stem cells (MSC) from wild-type FVB mice (Fig. 1A).

Line 133-135: As a result, after approximately three passages (~ 3 weeks), fibroblasts disappeared from the flasks, and the primary cultures exhibited a uniform morphology corresponding to cancer cells (Fig 1B).

Line 137-138: The established cell line, designated as CST, consists of adherent cells exhibiting fillopodial and lamellipodial structures related to mesenchymal morphology (Fig. 1C).

Line 138-140: Despite their mesenchymal phenotype, CST cells are of epithelial origin, as evidenced by the truncation of Brca1 gene that occurred in the mammary epithelium of K14cre;Brca1F/F;p53F/F mice (Fig. 1D).

Figure S1 A: in the table there is: “FVB mouse fat MSC”. Probably there should be: “FVB mouse bone marrow MSC”.